

# Relationship between healthcare seeking and pain expansion in patients with nonspecific chronic low back pain

Mónica Grande-Alonso[1,2], Daniel Muñoz-García[2], Ferran Cuenca-Martínez[1,2], Laura Delgado-Sanz[1], María Prieto-Aldana[2], Roy La Touche[1,2,3,4] and Alfonso Gil-Martínez[1,4,5]

[1] Departamento de Fisioterapia, Centro Superior de Estudios Universitarios La Salle, Universidad Autónoma de Madrid, Madrid, Spain
[2] Motion in Brains Research Group, Institute of Neuroscience and Sciences of the Movement (INCIMOV), Centro Superior de Estudios Universitarios La Salle, Universidad Autónoma de Madrid, Madrid, Spain
[3] Instituto de Neurociencia y Dolor Craneofacial (INDCRAN), Madrid, Spain
[4] Instituto de Investigación Sanitaria del Hospital Universitario La Paz (IdiPAZ), Madrid, Spain
[5] CranioSPain Research Group, Departamento de Fisioterapia, Centro Superior de Estudios Universitarios La Salle, Universidad Autónoma de Madrid, Madrid, Spain

Corresponding author
Roy La Touche,
roylatouche@lasallecampus.es,
roylatouche@yahoo.es

## ABSTRACT

**Objectives**. Low back pain (LBP) is the most prevalent musculoskeletal problem, which implies a high rate of chronicity. The chronicity of symptoms can lead to pain expansion. The main objective of this study was to assess whether there were differences between patients with nonspecific chronic LBP (CLBP) who sought healthcare compared to those who did not in terms of pain expansion.

**Methods**. Ninety individuals participated in the study and were divided into three groups: 30 patients who sought care; 30 patients who did not seek care; and 30 asymptomatic individuals. The primary variable analyzed was pain expansion. Secondary physical and psychological variables were assessed later, and a regression analysis was performed.

**Results**. Patients who sought help showed significant differences in pain expansion and pain intensity compared with the group who did not seek help, with a medium effect size (0.50–0.79). The regression model for the care-seeking group showed that dynamic balance with the left leg and depression were predictors of percentage pain surface area (34.6%). The combination of dynamic balance, range of movement in flexoextension and depression were predictors of widespread pain (48.5%).

**Conclusion**. Patients who soughtcare presented greater pain expansion than patients whodidnot. A combination of functional and psychological variables can significantly predict pain expansion in patients with nonspecific CLBP who seek help.

# INTRODUCTION

Low back pain (LBP) is the most prevalent musculoskeletal problem and the fourth pathological cause of disability, which implies a high rate of chronicity and absenteeism (*Hoy et al., 2012*). Between 13.5% and 47% of the general population has been found to

have chronic musculoskeletal pain (*Cimmino, Ferrone & Cutolo, 2011*). Chronic pain is defined as pain that persists beyond 6 months, leading to neural, somatic, cognitive and behavioral disturbances. Such alterations can lead to maladaptive neuroplastic changes at the medullary and supramedullary level (*Hashmi et al., 2013*; *Merskey & Bogduk, 1994*; *Walsh et al., 2008*), associated with the concept of nociplastic pain. Such pain is defined, according to the International Association for the Study of Pain, as "*pain that arises from altered nociception despite no clear evidence of actual or threatened tissue damage causing the activation of peripheral nociceptors or evidence for disease or lesion of the somatosensory system causing the pain*" (*IASP, 2017*).

Among the characteristics found in this population, it has been observed that individuals with chronic low back pain (CLBP) can present functional alterations, such as changes in dynamic stability or range of motion, physical alterations and somatosensory disturbances (*Sadler et al., 2017*; *Laird et al., 2014*; *Tong et al., 2017*). All these characteristics make CLBP a major public health problem, given it causes major disability and a reduction in quality of life of those who experience it. However, the prevalence of health system use in Europe for LBP has been shown to be low, at 48% (*Beyera, Brien & Campbell, 2019*). Thus, numerous studies have evaluated the factors that lead a patient with CLBP to seek professional help (*Buchan et al., 2016*; *IJzelenberg & Burdorf, 2004*; *Traeger et al., 2019*). Research has shown that there is a positive association between care seeking and sociodemographic variables such as age, as well as other variables related to the experience of pain, such as pain intensity, frequency of pain episodes, disability and chronicity of symptoms and cognitive variables (*Jacob, Zeev & Epstein, 2003a*; *Jacob, Zeev & Epstein, 2003b*; *Mannion, Wieser & Elfering, 2013*; *Szpalski et al., 1995*). Similarly, although some care-seeking patients have been shown to develop an external locus of control, high rates of disability and a greater influence of psychological factors in regard to their perception of pain, no studies have shown how these relate to care seeking for pain expansion (*Ferreira et al., 2010*; *Rollman et al., 2012*; *Rollman et al., 2013*).

CWP is defined as pain present on both sides of the body, involving an expansion to the upper limbs, the lower limbs and the axial skeleton (*Wolfe et al., 1990*). CWP is an important clinical variable because it provides us with abundant information. This condition has been shown to have a high impact on functionality, quality of life and psychological factors (*Landmark et al., 2013*; *Papageorgiou, Silman & Macfarlane, 2002*). *Reis et al. (2018)* had studied brain regions related to emotions and cognition in patients with chronic pain and found that changes in brain function are related to CWP in various regions of the body. In particular, CWP was associated with high rates of anxiety and depression, but evidence is still scarce (*Hagen et al., 2011*; *Ris et al., 2019*).

The main objective of this study was to assess whether there were differences between patients with nonspecific CLBP who sought health care compared with those who did not, in terms of pain expansion. The secondary objective was to determine which factors predictive of greater pain expansion are present in patients with nonspecific CLBP based on their search for care.

## MATERIALS & METHODS

### Design and sample

This study was cross-sectional with a non-probabilistic sample, with the aim of assessing somatosensory, physical and psychosocial variables in patients with nonspecific CLBP who seek care or not and asymptomatic individuals. The trial was conducted in accordance with the Strengthening the Reporting of Observational Studies in Epidemiology statement (*Von Elm et al., 2008*). Following the Helsinki Declaration, the Ethics Committee of the La Paz Hospital in Madrid, Spain approved our study (PI-2567) for clinical research in a public reference hospital in Madrid (Spain), and written informed consent was obtained from all participants.

The participants were recruited between April 2017 and January 2018. The sample was recruited from our university campus and the local community through flyers, posters, social media and outpatients of a primary healthcare center in Madrid, Spain.

A consecutive nonprobabilistic convenience sample of 90 individuals was recruited. Participants were classified into the following groups: group 1 was composed of 30 patients with nonspecific CLBP who did not seek care; group 2 was composed of 30 patients with nonspecific CLBP who sought care; and group 3 was composed of 30 asymptomatic individuals. The symptomatic participants were assigned to one group or the other according to whether they sought care from a health professional for their musculoskeletal condition. Those patients who at no time went to any health professional for their problem were classified in the non-care-seeking group. These patients were recruited via flyers placed at the university center and in the local community. The patients who had gone to a primary care doctor due to the presence of CLBP were assigned to the care-seeking group, given they had an intention to be treated.

Patients with nonspecific CLBP were selected if they met the inclusion criteria defined by the National Institute for Health and Care Excellence in the LBP Guidelines defines the nature of LBP as "Tension, soreness and/or stiffness in the lower back region for which it is not possible to identify a specific cause of the pain. Several structures in the back, including the joints, discs and connective tissues, may contribute to symptoms" (*Savigny, Watson & Underwood, 2009*). The following inclusion criteria were also considered on the basis of a previous investigation (*Grande-Alonso et al., 2019*): (a) LBP for at least the prior 3 months; (b) LBP of a nonspecific nature; (c) men and women aged 18 to 65 years (*Carmona et al., 2001*); (d) LBP for at least 10 days per month (*Goubert, Danneels & Graven-nielsen, 2017*); the time between seeking care and recruitment was 5–7 days; and (e) an intensity of pain between 3 and 10 on the visual analogue scale (VAS).

Individuals were excluded if they met any of the following exclusion criteria: (a) comorbidities, such as the presence of neurological signs (e.g., weakness perceived in the lower limbs), systemic rheumatic disease (including fibromyalgia) or central nervous system disease; (b) the presence of psychiatric diagnosis or severe cognitive impairment; (c) illiteracy; (d) understanding or communication difficulties; and (e) insufficient Spanish language comprehension to follow measurement instructions.

Finally, asymptomatic individuals were excluded if they had a history of spinal pain, another condition of chronic pain or had a diagnosis of any systemic disease.

## Procedure

After consenting to participate, all the participants received a sociodemographic questionnaire to complete on the day of the measurement, which collected sex, date of birth and educational level. Next, each participant completed a set of self-report measures, and we evaluated the pain drawings in both the care-seeking and non-care-seeking groups.

Next, the evaluator conducted a semi-structured interview with each of the patients in which questions were asked about their symptomatology (e.g., intensity, frequency and severity of symptoms), demographics and certain questions to determine whether they were in search of treatment, based on previous literature (*Macfarlane et al., 2003*; *Rollman et al., 2013*). Once the questionnaires were completed, the participant painted the regions of pain on the body chart. Then, according to the procedure of Dos Reis et al., we used an electronically scanned version of the body diagram and open-source software to calculate the total body area in each pain diagram (*Dos Reis et al., 2016*). Based on the literature, we decided to calculate the percentage pain surface area (PPSA) and count the number of pain sites in order to evaluate widespread pain (WP) (*Dragioti et al., 2017*; *Hägg et al., 2003*; *Persson, Garametsos & Pedersen, 2011*; *Visser et al., 2014*). A previous study of patients with chronic pain (*Muñoz García et al., 2016*) had shown that both PPSA and WP measures were helpful when assessing pain behavior. The PPSA shows the percentage surface area of pain and the WP calculates how widespread that magnitude is (in number of sites) over the body surface area.

Finally, a physiotherapist instructed the patients regarding the physical test to be performed, and they were supervised during the session. The first test that was performed was an evaluation of the range of movement in flexoextension (ROMFE) and lateral flexion movements (ROMLF). The protocol for measuring range of motion consisted of the following process: the patient was placed in a standing position with arms along the body; the physiotherapist marked the spinous process of T12 and S2 to place the mobile device; the patient was then asked to perform a maximum trunk flexion (*Bedekar et al., 2014*), followed by a maximum trunk extension. Three measurements were taken, and the average of the differences presented between the two reference points was calculated. Then, the physiotherapist added the degrees of flexion and extension movements. Finally, the patient was placed in the same position with the mobile device placed on T12, the movement of the complete lateral flexion being evaluated (*Bedekar et al., 2014*).

The final test that was performed was the evaluation of dynamic balance by means of the Y Balance Test (YBT). This test was performed in a single-limb stance while simultaneously moving the nonstanding limb in 3 different directions: anteriorly, posteromedially and posterolaterally (*Plisky et al., 2009*; *Teyhen et al., 2014*). The composite reach distance (%) was calculated by the sum of the 3 reach directions divided by 3 times the limb length per 100 (*Shaffer et al., 2013*). These measurements were obtained for the right and left sides.

## Outcome Measures
### Primary variable
*Pain drawings.* Pain drawings (PPSA and WP) were assessed using open-source software to calculate the total body area in each pain diagram (*Dos Reis et al., 2016*), which has been shown to have good intrarater reliability with an intraclass correlation coefficient (ICC) = 0.99; 95% CI [0.98–0.99] $P < 0.001$. The inter-rater reliability for the measurement was ICC = 0.989; 95% CI [0.980–0.994]; $P < 0.001$ (*Dos Reis et al., 2016*).

### Secondary variables
*Pain intensity.* Self-reported pain was assessed using the Spanish version of the VAS. The VAS is a 100-mm line with 2 endpoints representing the extreme states, "no pain" and "pain as bad as it could be". It has been shown to have good retest reliability ($r = 0.94$, $P > 0.001$) (*Bijur, Silver & Gallagher, 2001*).

*Frequency of pain.* The frequency of pain was evaluated based on the number of days with pain during the last month (*Grande-Alonso et al., 2019*).

*Frequency of medication.* The frequency of medication was evaluated based on the number of days the patient had taken medication for LBP in the last month (*Grande-Alonso et al., 2019*).

*Level of physical activity.* The physical activity (PA) level was measured using the International Physical Activity Questionnaire in its short version. It consists of 9 items that quantify the time that the individual devotes to perform any PA of vigorous or moderate intensity. This questionnaire presents an ICC of 0.76 (95% CI) (*Craig et al., 2003*).

*Range of motion.* The range of motion was evaluated with a digital inclinometer based on the mobile application, iHandy. It has been shown to have good intrarater and inter-rater reliability, with ICC over 0.80 (95% CI) (*Kolber et al., 2013*).

*Dynamic balance.* Dynamic balance was measured using YBT. The YBT has shown good to excellent intrarater (0.85–0.91) and inter-rater (0.99–1.00) reliability (*Plisky et al., 2009*).

*Anxiety and depression.* Aanxiety and depression levels were assessed using the Hospital Anxiety and Depression Scale (HADS). The scale has two subscales of 7 items each that measure anxiety and depression (*De Las Cuevas-Castresana, García-Estrada Pérez & Gónzalez de Rivera, 1995*). The HADS presented an internal consistency (Cronbach's alpha) from 0.80 to 0.93 for the anxiety and 0.81 to 0.90 for the depression subscales (*Herrmann, 1997*).

*Fear of movement.* Fear of movement was assessed using the 11-item Spanish version of the Tampa Scale of Kinesiophobia (TSK-11), which has a Cronbach's alpha of 0.78 (*Gómez-Pérez, López-Martínez & Ruiz-Párraga, 2011*). The final score can range between 11 and 44 points, with higher scores indicating greater perceived fear of movement.

*Low back disability.* Physical disability due to LBP was assessed using the Spanish version of the Roland-Morris Disability Questionnaire (RMDQ), which presented an internal consistency (Cronbach's alpha) of 0.84 to 0.93 and test–retest reliability ranging between 0.72 and 0.91 (*Kovacs et al., 2002*; *Roland & Fairbank, 2000*).

## Sample Size

We conducted a pilot study to determine the effect size between non-care seekers with nonspecific CLBP and care seekers with nonspecific CLBP using a pain drawing. The pilot study included 15 patients from each group and obtained an effect size (Cohen's *d*) of 0.66. The sample size was estimated with G\*Power 3.1.7 for Windows (G\*Power from University of Dusseldorf, Germany) (*Faul et al., 2007*). We opted to use an independent *t*-test in order to detect differences between both symptomatic groups for WP. Moreover, we used an alpha error level of 0.05, a statistical power of 80% (1-B error), and an effect size of 0.66. A total sample size of 60 patients (30 non-care seekers with nonspecific CLBP and 30 care seekers with nonspecific CLBP) was estimated to ensure reliability.

## Data analysis

The sociodemographic and clinical variables of the participants were analyzed. The data were summarized using frequency counts, descriptive statistics, summary tables and figures.

The data analysis was performed using the Statistics Package for Social Science (SPSS 20.00, IBM Inc., USA). The categorical variables are shown as frequency and percentage. The quantitative results of the study are represented by descriptive statistics (CI, mean, and standard deviation). For all variables, the z-score was assumed to follow a normal distribution based on the central limit theorem, given the groups had more than 30 participants (*Kwak & Kim, 2017*; *Mouri, 2013*; *Nixon, Wonderling & Grieve, 2010*). Student's t- test was used for the nonspecific CLBP group comparisons (months of pain, pain intensity, days of pain/month, days of medication/month, PPSA, WP and RMDQ). Cohen's *d* effect sizes were calculated for a *post hoc* analysis of the outcome variables. According to Cohen's method, the magnitude of the effect was classified as small (0.20–0.49), medium (0.50–0.79) or large (0.80).

A one-way analysis of variance (ANOVA) was used to analyze numerical variables among the asymptomatic participants, the care-seeking group and the non-care-seeking group (sociodemographic variables, ROMFE, ROMLF, CRDL, CRDR, HAD_D, HAD_A and TSK-11). Significant ANOVA findings were followed up using a *post hoc* test with Bonferroni correction. We calculated the partial eta-squared ($\eta_p^2$) as a measurement of the effect size for each main effect and interaction in the ANOVAS. For this analysis, 0.010–0.059, 0.060–0.139, and >0.14 represented small, medium and large effects, respectively (*Cohen, 1988*; *Cohen, 1973*).

To ensure control of the false positive rate, an adjustment of the *p*-values using the Benjamini–Hochberg method was used (false discovery rate [FDR]). The FDR correction ensures that in no case, if working with 95% confidence, will there be more than 5% of variables in which a false positive has occurred (*Benjamini & Hochberg, 1995*).

We examined the PPSA and WP associations with psychological, functional and somatosensory measures, using Pearson's correlation coefficient. A Pearson correlation

coefficient >0.60, 0.30–0.60 and <0.30 indicated high, medium and low correlations, respectively (*Hinkle, Wiersma & Jurs, 1990*). The variables that showed the highest correlation with PPSA and WP were used to perform the subsequent regression analysis.

A multiple linear regression analysis was performed to estimate the strength of the associations between the PPSA and WP results. PPSA and WP variables were used as predictors. Considering the variables more strongly correlated with PPSA and WP, we performed the linear regression analysis. The strength of the association was examined using regression coefficients (B), $P$ values and adjusted $R^2$. Standardized beta coefficients were reported for each predictor variable. We included the final reduced models to allow a direct comparison between the predictor variable and the criterion variable, which we studied. For the data analysis, we used a 95% CI and a $P$ value of less than 0.05.

## RESULTS

A total of 90 participants completed the study (30 patients with nonspecific CLBP who sought care, 30 patients with nonspecific CLBP who do not seek care and 30 asymptomatic controls). Table 1 shows the sociodemographic characteristics of the study participants.

### Primary variable
#### *Pain expansion*
Statistically significant differences were observed in the pain drawing, with pain expansion greater in care seekers (Figs. 1 and 2). Student's $t$-test (for independent samples) revealed significant differences between the groups for PPSA ($t = -2.50$; $q = 0.02$; $d = 0.64$) and WP ($t = -2.77$; $q = 0.02$; $d = 0.71$). Table 2 shows the intergroup comparison. A bar graph was made based on the search for help and the variables of the pain expansion. This graph shows that there was little dispersion in the sample with respect to these variables (Figs. 3 and 4).

### Secondary Variables
Regarding the secondary variables, the strongest results showed significant differences in pain intensity ($t = -2.35$; $q = 0.03$; $d = -0.73$) and frequency of medication ($t = -2.86$; $q = 0.003$; $d = -0.73$) between care seekers and non-care seekers, with higher intensity in care seekers. Data with respect to physical, somatosensory and psychosocial variables are summarized in Tables 2 and 3.

### Correlation and Regression Analyses
Pearson's correlation analysis showed only a moderate correlation, observed in the pain drawing, between functional and psychological variables in the care-seeking group of patients with nonspecific CLBP. The most significant correlations in this group were the association between right and left dynamic stability and the number of pain sites ($r = -.540$, $P < 0.01$; $r = -.564$, $P < 0.01$, respectively), the association between depression and the number of pain sites ($r = .436$, $P < 0.05$) and the association of the same variable with PPSA ($r = .428$, $P < 0.05$). Finally, a negative correlation was also established between the ROMFE and the PPSA ($r = -.391$, $P < 0.05$) and between the days of medication intake and the number of pain sites ($r = .393$, $P < 0.05$) (Table 4).

**Table 1** Descriptive statistics of socio-demographic data.

| Measures | Asymptomatic (n = 30) | Non care seekers (n = 30) | Care seekers (n = 30) | FDR-Adjusted p-values a.k.a q value |
|---|---|---|---|---|
| Age[a] | 38.30 ± 13.09 | 37.93 ± 12.30 | 46.07 ± 10.90 | .02[**] |
| Gender[b] | | | | .87 |
| Male (%) | 13 (43.3) | 14 (46.7) | 12 (40) | |
| Female (%) | 17 (56.7) | 16 (53.3) | 18 (60) | |
| Height (cm)[a] | 167.47 ± 9.76 | 171 ± 8.42 | 166.73 ± 10.14 | .23 |
| Weight (kg)[a] | 69.20 ± 16.87 | 72.22 ± 11.95 | 73.54 ± 17.50 | .60 |
| BMI (Kg/m$^2$) | 24.48 ±.83 | 24.61 ±.55 | 26.4 ± 1.05 | .23 |
| Educational Level[b] | | | | .09 |
| Primary Education (%) | 4 (13.3) | 3 (10) | 10 (33.3) | |
| Secondary education (%) | 4 (13.3) | 2 (6.6) | 5 (16.7) | |
| College education (%) | 22 (73.3) | 25 (83.3) | 15 (50) | |
| Level of physical activity[b] | | | | .03[**] |
| Mild (%) | 5 (16.7) | 4 (13.3) | 11 (36.7) | |
| Moderate (%) | 8 (26.7) | 15 (50) | 12 (40) | |
| Vigorous (%) | 17 (56.7) | 11 (36.7) | 7 (23.3) | |
| Medication[b] | | | | .003[**] |
| None | 30 (100.0) | 19 (63.3) | 7 (23.3) | |
| Ibuprofen | 0 (0.0) | 10 (33.3) | 20 (66.7) | |
| Paracetamol | 0 (0.0) | 1 (3.3) | 3 (10.0) | |

**Notes.**

Values are presented as mean ± standard deviation or number (%).

BMI, Body Mass Index.

[b]Chi-square test.

[a]One-way analysis of variance q value.

False Discovery Rate Correction.

[*]q < .05.

[**]q < .01.

In contrast, in the group of patients with nonspecific CLBP who did not seek care, we found no correlation between the main variable of the study and variables of a functional or psychological nature (Table 4).

The regression models for the criteria variables (PPSA and WP) are presented in Table 5. The regression model for the nonspecific CLBP care-seeking group showed that a combination of CRD L (%) and depression were predictors of PPSA (34.6% of variance). The variables of CRD R (%) and days of medication per month were excluded from the analysis. Instead of the combination of CRD L, ROMFE and depression were predictors of WP (48.5% of variance). The variable of CRD R (%) was excluded from the analysis. For the non-care seeking group with nonspecific CLBP, the regression analysis was not performed because no significant correlation was found with the main study variable.

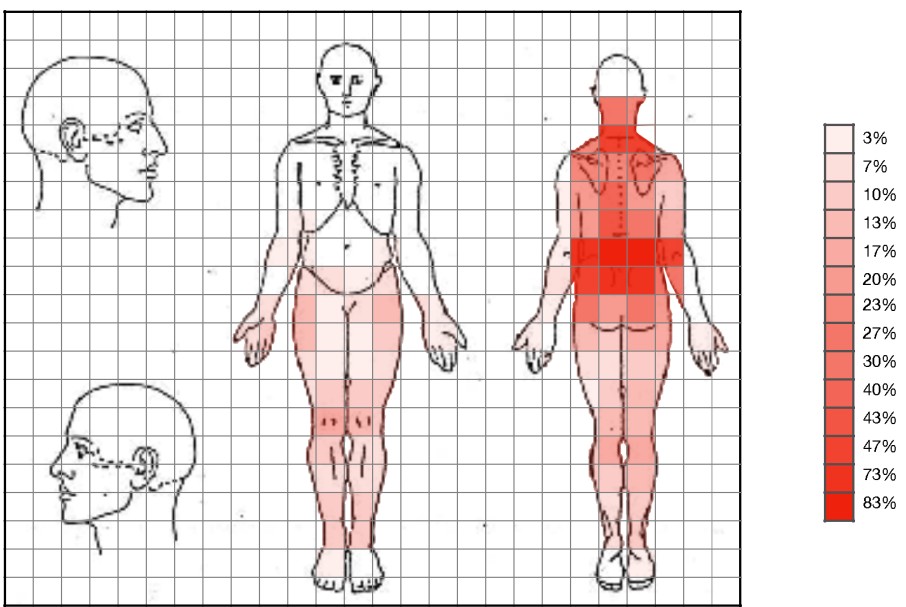

**Figure 1** Expansion of pain in patients who sought care.

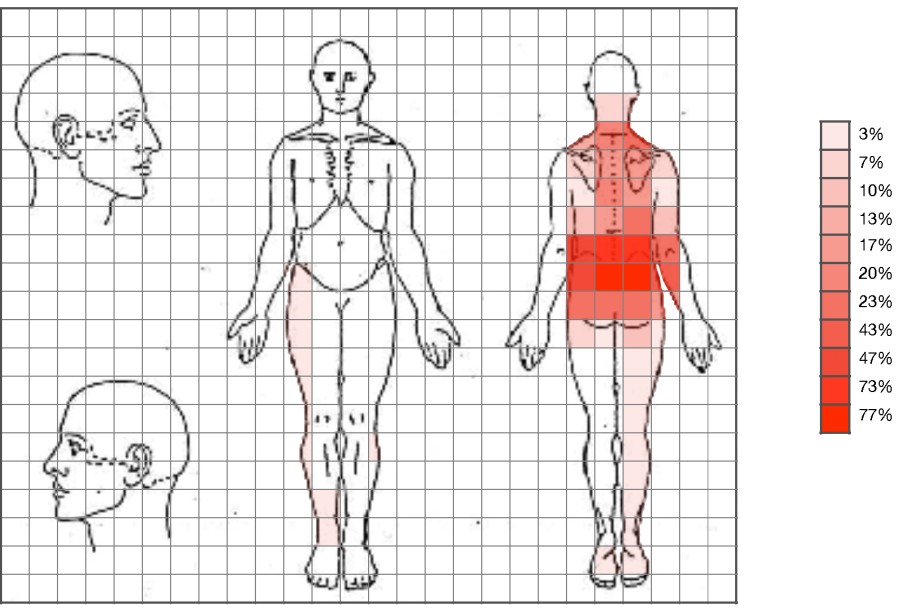

**Figure 2** Expansion of pain in patients who did not seek care.

**Table 2  Descriptive and multiple comparisons of somatosensory and motor variables.**

| Measures | Mean ± SD | | | Difference of means (95% CI); Effect size ($d$) | FDR-Adjusted |
|---|---|---|---|---|---|
| | Aymptomatic ($n = 30$) | Non care seeker ($n = 30$) | Care seekers ($n = 30$) | a) Non care seekers vs Care seekers<br>b) Non care seekers vs asymptomatic<br>c) Care seekers vs asymptomatic | $p$-value a.k.a $q$ value |
| Months of pain[b] | – | 69.47 ± 56.33 | 77.70 ± 119.30 | a) −8.23 (−56.45; 39.98) $d = -.08$<br>b) –<br>c) – | .77 |
| Pain intensity (VAS)[b] | – | 48.13 ± 16.55 | 58.27 ± 16.81 | a) 10.13[*] (−18.75; −1.51) $d = -.60$<br>b) –<br>c) – | .03[*] |
| Days of pain /month[b] | – | 20.40 ± 7.06 | 23.60 ± 7.49 | a) 3.20 (−6.96; .56) $d = -.43$<br>b) –<br>c) – | .12 |
| Days of medication /month[b] | – | 3.10 ± 7.63 | 9.57 ± 9.74 | a) 6.46[**] (−10.99; −1.94) $d = -.73$<br>b) –<br>c)– | .003[**] |
| PPSA[b] | – | 2.86 ± 2.72 | 6.67 ± 7.88 | a) 3.81[*] (−6.86; −.76) $d = -.64$<br>b)–<br>c) | .02[*] |
| WP[b] | – | 9.03 ± 7.33 | 17.63 ± 15.28 | a) 8.60* (−14.79; −2.40) $d = -.71$<br>b) –<br>c) – | .02[*] |
| ROMFE[a] | 75.05 ± 11.86 | 65.44 ± 9.01 | 55.81 ± 9.11 | a) 9.63[**] (3.21; 16.05) $d = 1.06$<br>b) 9.60[**] (3.18; 16.02) $d = -.91$<br>c) 19.24[**] (12.87; 25.60) $d = -1.81$ | <.001[**] |
| ROMLF[a] | 58.82 ± 8.25 | 52.83 ± 13.85 | 45.37 ± 7.80 | a) 7.46* (.94; 13.98) $d = .66$<br>b) 5.98 (−.53; 12.50) $d = -.52$<br>c) 13.44[**] (6.92; 19.96) $d = -1.67$ | <.001[**] |
| CRD L (%)[a] | 108.12 ± 8.27 | 94.53 ± 13.07 | 87.18 ± 13.83 | a) 7.34 (−.20; 14.90) $d = .54$<br>b) 13.59[**] (−21.15; −6.04) $d = -1.24$<br>c) 20.94[**] (−28.49; −13.39) $d = -1.83$ | <.001[**] |
| CRD R (%)[a] | 108.24 ± 8.63 | 93.85 ± 13.38 | 87.46 ± 13.21 | a) 6.38 (−1.14; 13.91) $d = .63$<br>b) 14.39[**] (−21.92; −6.86) $d = -1.10$<br>c) 20.78[**] (−28.31; −13.24) $d = -1.86$ | <.001[**] |

**Notes.**

Values are presented as mean ± standard deviation.

VAS, Visual Analogue Scale; PPSA, Percentage Pain Surface Area; WP, Widespread Pain; RMDQ, Roland-Morris Disability Questionnaire; TSK-11, Tampa Scale of Kinesiophobia; HAD_D, Depression; HAD_A, Anxiety; HAD_A, Range of movement in Flexoextension; ROMLF, Range of movement in lateral flexion; CRD L (%), Composite Reach Distance Left (%); CRD R (%), Composite Reach Distance Right (%).

[a]One-way analysis of variance.
[b]Independent Student's t test.
q value; False Discovery Rate Correction.
*$q < .05$.
**$q < .01$.

# DISCUSSION

## Pain expansion and care seeking

The main differences between the patient groups analyzed in this study concerned pain expansion. Individuals who sought care had almost double the number of regions affected by pain than those who did not seek care.

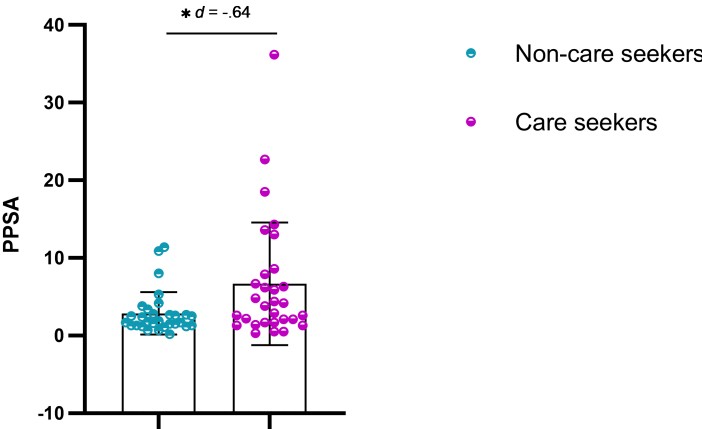

**Figure 3** **Scatter plot for PPSA.** $q < 0.05$.

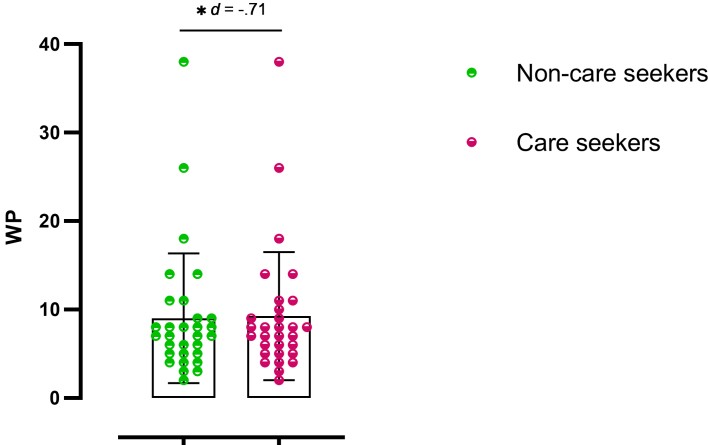

**Figure 4** **Scatter plot for WP.** $q < 0.05$.

The body pain diagram is a tool that provides relevant information. Its use has also been considered for assessing the psychological state of patients due to the greater the number of pain areas and the greater impact that psychological factors exert on the clinical condition (*Haefeli & Elfering, 2006*). In our study, individuals with greater pain expansion sought care and presented significantly greater anxiety and depression, with a medium effect size. These data agree with previous studies, in which patients with chronic pain in several body regions at the same time had anxiety levels that were considered pathological compared with patients having more localized pain. Moreover, they had greater depression and feelings of distress (*Abbott et al., 2015*; *Muñoz García et al., 2016*). Evidence has shown that CWP not only correlates with psychological variables but also correlates with somatosensory variables, such as the severity of symptoms or mechanical hyperalgesia (*Ferrer-Peña et al., 2018*). Our results showed that patients with greater pain expansion who were in search of help presented a greater alteration in pain intensity and

**Table 3  Descriptive and multiple comparisons of psychological variables.**

| Measures | Mean ± SD | | | Difference of means (95% CI); Effect size ($d$) | FDR-Adjusted |
| --- | --- | --- | --- | --- | --- |
| | Ay|mptomatic ($n = 30$) | Non care seekers ($n = 30$) | Care seekers ($n = 30$) | a) Non care seekers vs Care seekers<br>b) Non care seekers vs asymptomatic<br>c) Care seekers vs asymptomatic | p-value<br>a.k.a q-value |
| RMDQ[b] | – | 3.97 ± 2.32 | 7.73 ± 4.44 | a) 3.76[**] (−5.60; −1.93) $d = -1.06$<br>b) –<br>c) – | .003[**] |
| HAD_A[a] | 5.10 ± 3.52 | 5.43 ± 3.51 | 8.10 ± 4.52 | a) 2.66[*] (.22; 5.12) $d = -.65$<br>b) .33 (−2.12; 2.78) $d = .09$<br>c) 3.00[**] (.55; 5.45) $d = .74$ | .02[*] |
| HAD_D[a] | 1.67 ± 2.35 | 2.37 ± 2.41 | 4.77 ± 4.01 | a) 2.40[**] (.49; 4.31) $d = -.72$<br>b) .70 (−1.21; 2.61) $d = .29$<br>c) 3.10[**] (1.19; 5.01) $d = .94$ | <.001[**] |
| TSK-11[a] | 20.33 ± 6.45 | 24.27 ± 5.86 | 29.33 ± 5.51 | a) 5.06[**] (1.31; 8.82) $d = -.88$<br>b) 3.93[*] (.18; 7.69) $d = .63$<br>c) 9.00[**] (5.24; 12.76) $d = 1.50$ | <.001[**] |

**Notes.**

Values are presented as mean ± standard deviation.

RMDQ, Roland-Morris Disability Questionnaire; HAD_A, Anxiety; HAD_D, Depression; TSK-11, Tampa Scale of Kinesiophobia.

[a]One-way analysis of variance.

[b]Independent Student's t test.

q value; False Discovery Rate Correction.

[*]$q < .05$.

[**]$q < .01$.

in medication intake per month. Pain expansion and the presence of generalized allodynia include as an underlying neurophysiological mechanism a possible central sensitization process (*Lluch-Girbés et al., 2016*). This process might explain why our patients seeking help had greater pain expansion, and thus, greater pain intensity, longer symptom duration and a greater involvement of psychological variables.

Along these lines, our results found that those patients with greater pain expansion who also sought care presented a significantly greater restriction of movement compared with the other groups. Other studies have shown that individuals with LBP had greater restriction of movement and slower speed of execution compared with asymptomatic individuals (*Laird et al., 2014*; *Sadler et al., 2017*). This reduction in range of motion and speed of execution are correlated with variables of a psychological nature, such as fear of movement (*Thomas et al., 2008*). These results are in line with the results obtained by Landmark, which show that the presence of widespread pain implies an alteration in functionality (*Landmark et al., 2013*).

In terms of dynamic stability, our results demonstrated significant differences between both symptomatic groups with respect to the asymptomatic group. The evidence available thus far shows that dynamic balance is reduced in patients with LBP (*Hooper et al., 2016*). Moreover, there were no differences based on their search for care. A recent study on patients with chronic hip pain had also shown that the dynamic stability in this population was altered and correlated with cognitive and sensory variables (*Ferrer-Peña, Moreno-López & Calvo-Lobo, 2018*).

**Table 4  Pearson correlation coefficient for all variables in patients group.**

| | Months of pain | Days of pain /month | Days of medication /month | VAS | PPSA | WP | RMQD | TSK-11 | HAD_D | HAD_A | TPD | ROMFE | ROMLF | CRD R (%) | CRD L (%) | GROUP |
|---|---|---|---|---|---|---|---|---|---|---|---|---|---|---|---|---|
| PPSA | .091 | .298 | .274 | .142 | 1 | .925[b] | .092 | .268 | .428[a] | .072 | .259 | −.391[a] | −.288 | −.488[b] | −.498[b] | **Care seekers** |
| | .161 | .222 | .176 | .137 | 1 | .909[b] | .028 | .012 | .262 | .157 | .037 | −.159 | −.240 | −.001 | −.009 | **Non care seekers** |
| WP | .007 | .120 | .393[a] | .112 | .925[b] | 1 | .104 | .163 | .436[a] | .191 | .286 | −.352 | −.280 | −.540[b] | −.564[b] | **Care seekers** |
| | .219 | .209 | .161 | .218 | .909[b] | 1 | .182 | .000 | .282 | .191 | .073 | −.081 | −.312 | −.126 | −.110 | **Non care seekers** |

Notes.

VAS, Visual Analogue Scale; PPSA, Percentage Pain Surface Area; WP, Widespread Pain; RMDQ, Roland-Morris Disability Questionnaire; TSK-11, Tampa Scale of Kinesiophobia; HAD_D, Depression; HAD_A, Anxiety; HAD_A, Range of movement in Flexoextension; ROMLF, Range of movement in lateral flexion; CRD L (%), Composite Reach Distance Left (%); CRD R (%), Composite Reach Distance Right (%).

[a]$p < .05$.
[b]$p < .01$.

**Table 5** Regression model for PPSA and WP in care seekers group with non-specific CLBP.

| Care seekers | Overall model | | | | |
|---|---|---|---|---|---|
| PPSA | $R^2 = .391$, adjusted $R^2 = .346$, $F = 8.67$ | | | | |
| | **Predictor variables** | **Regression coefficient (B)** | **Standardized coefficient ($\beta$)** | **P-value** | **VIF** |
| | CRD L (%) | −.261 | −.458 | .005[b] | 1.01 |
| | HAD_D | .747 | .297 | .018[a] | 1.01 |
| | **Excluded variables** | | | | |
| | CRD R (%) | | .011 | .980 | 7.69 |
| | Days of medication /month | | .230 | .133 | 1.02 |
| WP | $R^2 = .502$, adjusted $R^2 = .485$, $F = 28.60$ | | | | |
| | CRD L (%) | −.325 | −.394 | <.001[b] | 1.51 |
| | ROM FE | −.187 | −.197 | .037[a] | 1.47 |
| | HAD_D | 1.27 | .347 | <.001[b] | 1,08 |
| | **Excluded variables** | | | | |
| | CRD R (%) | | −.047 | .885 | 17.92 |

**Notes.**

CLBP, Chronic Low Back Pain; VIF, Variance Inflation Factor; PPSA, Percentage Pain Surface Area; WP, Widespread Pain; HAD_D, Depression; ROM FE, Range of movement in Flexoextension; CRD L (%), Composite Reach Distance Left (%); CRD R (%), Composite Reach Distance Right (%).

[a] $p < .05$.
[b] $p < .01$.

### Characteristics of patients seeking care

It is important to define, taking into account the results obtained, which factors can influence the search for care in patients with CLBP. Numerous studies have determined that the rate of transition from non-CWP to CWP varies from 9% to 25% (*Jones et al., 2011*; *Larsson et al., 2012*; *McBeth, Lacey & Wilkie, 2014*; *Viniol et al., 2015*). A recent study published in 2019 had determined that the rate of CWP development in a period of 1 year in the general population is 5% and that the factors that most contribute to this development are chronicity, severity of symptoms and the involvement of at least 2 of 5 total body regions in which the interpretation of the body chart is divided (*Landmark et al., 2019*). Thus, given our results indicate that there are statistically significant differences in pain expansion between seekers and nonseekers of care in terms of age, pain intensity and disability, we find these patient characteristics relevant.

The most recent systematic review determined that due to the current methodological heterogeneity, it is difficult to clearly determine which social factors might lead to the search for health care. Even so, it is estimated that the prevalence rate of health system use by LBP in Europe is lower than in other countries such as the United States (*Beyera, Brien & Campbell, 2019*). In relation to the age differences, it is estimated that the prevalence of CLBP increases gradually from the age of 30, which could contribute to a greater demand for health care at a later age (*Hoy et al., 2010*). Research studies have found a positive association between age and medical service use, which could justify the significant difference between groups that we started with in this study (*Jacob, Zeev & Epstein, 2003a*; *Ono et al., 2015*; *Szpalski et al., 1995*). In contrast, other sociodemographic characteristics, such as marital status, have not shown any significant association with the use of the health system (*Ono et al., 2015*).

On the other hand, certain characteristics of the pain experience, such as pain intensity, are factors related to the search for health care (*Beyera, Brien & Campbell, 2019*). Given this agrees with the results observed in this study, a number of sociodemographic and pain-related factors could contribute to the search for health care.

Similarly, one study on patients with CLBP determined that impaired functionality and disability are factors positively associated with care seeking (*Ferreira et al., 2010*). Therefore, taking into account the studies on patients with temporomandibular dysfunction, care seeking could imply the presence of passive coping strategies. These strategies have shown a correlation with greater pain intensity, less functionality and higher rates of disability (*Alhowimel et al., 2018*; *Du et al., 2017*; *Jackson et al., 2014*; *Knittle et al., 2011*).

Psychological factors also have a great influence on patients with chronic pain; the most influential are fear of movement, anxiety and depression. The results of this study showed significant differences between all study groups; fear of movement was the most notable difference between individuals with CLBP seeking help and asymptomatic individuals, with a large effect size. In relation to our results, one study had shown that in patients with temporomandibular dysfunction, the decision to seek care was correlated with higher pain intensity and greater fear of movement (*Rollman et al., 2012*). This outcome is in line with our results, given the patients with nonspecific CLBP who sought health care had higher fear of movement rates than those who did not seek help, with a large effect size.

These results could also be due to a statistically significant difference in the level of physical activity (PA) between groups, which can contribute to an alteration of the pain inhibitory system; however, it has also been observed that patients with chronic pain who perform PA can improve their symptomatology and physical function (*Geneen et al., 2017*). Even so, some studies have shown no direct relationship between PA levels and pain intensity; instead, the level of PA could be correlated with functionality (*Griffin, Harmon & Kennedy, 2012*; *Hendrick & Hale, 2011*). Along these lines, a recent study in patients with nonspecific CLBP who performed PA despite their pain found that, despite their pain, they showed no differences in range of motion and dynamic stability compared with asymptomatic individuals. Therefore, this could be a reason why those who seek help who also have a lower level of PA have statistically significant differences, with greater effect size, compared with asymptomatic individuals in those functional variables (*Nieto-Garcia et al., 2019*).

### Predictive factors for pain expansion

In contrast, the combination of psychological variables such as fear of movement and disability as well as pain intensity, predicted dynamic stability in 43.8% of the individuals with chronic hip pain (*Ferrer-Peña et al., 2018*). These data are important, given the relationship between CLBP and hip pain, especially considering that in our study, dynamic stability together with depression and/or ROMFE were variables predictive of pain expansion).

Finally, taking into account the prognostic factors in CLBP, the presence of depression and anxiety influence the maintenance and recurrence of symptoms (*Castro et al., 2011*; *Croft, Dunn & Raspe, 2006*). A study by Hung et al. had found that depression was the most

powerful psychological factor related to disability (*Hung, Liu & Fu, 2015*). On the other hand, there is little evidence on possible predictors of pain expansion. Our study shows that variables such as dynamic stability, in combination with depression and/or ROMFE, can predict pain expansion in 34.6% and 48.5% (PPSA and WP, respectively) of those who sought care. It is important to note that pain expansion is a powerful predictor of an alteration in the modulation of pain, in addition to the influence of variables such as anxiety, low expectation of recovery and hypersensitivity in various musculoskeletal pain conditions (*Clark et al., 2017*; *Smart et al., 2010*). On the other hand, among the predictors of pain expansion in patients with CLBP are the presence of psychosomatic symptoms, the female sex and a long duration of symptoms (*Viniol et al., 2015*).

### Study limitations

This study has several limitations. First, the patients seeking help in this study presented a lower level of PA compared with asymptomatic individuals and with patients with nonspecific CLBP who did not seek help. A systematic review had determined that the evidence is limited in terms of improvement in pain severity as a result of exercise, instead determining that there is an improvement in physical and psychological variables through this intervention (*Geneen et al., 2017*). Another important limitation is age, given there were significant differences between groups. However, research studies have shown that there is a positive correlation between age and the search for health care (*Jacob, Zeev & Epstein, 2003b*; *Ono et al., 2015*; *Szpalski et al., 1995*). Although previous research on pain expansion had also found statistically significant differences in terms of the age variable, the difference was not considered clinically relevant (*Muñoz García et al., 2016*).

We recommend a more exhaustive evaluation based on somatosensory variables (pressure pain thresholds, thermal thresholds and temporal summation) in the future to determine the presence of a central sensitization process.

Another limitation is the selection of patients. Individuals with a medical diagnosis of fibromyalgia were excluded. It is possible that there were some cases with similar characteristics to this pathology, which should be taken into account. Schaefer et al. had concluded that future studies are necessary, to determine the presence of fibromyalgia or CWP, given errors can be made in sample selection due to the fact that these are different clinical conditions according to the International Classification of Functioning, Disability and Health (*Schaefer et al., 2016*).

One of the main limitations of this study is the possibility of a type I error, which has been solved by adjusting the *p*-values using the Benjamini–Hochberg method (false discovery rate).

Finally, the results of the present study should be interpreted with caution, given it is a cross-sectional study; thus, causal relationships cannot be established.

## CONCLUSIONS

The results of this study show that patients who sought care presented greater WP than patients who did not seek care. In addition, psychological and disability factors have a greater influence on the experience of pain in patients with nonspecific CLBP who seek

care, as well as their having a reduced range of motion and less ability to discriminate two points. The CRD L, depression and days of medication per month were covariates of PPSA (34.6% of variance), and the composite reach distance percentage of the left leg, depression and ROMFE were covariates of WP (48.5% of variance) for patients with nonspecific CLBP who sought care.

## ACKNOWLEDGEMENTS

The authors thank the Miraflores Health Center (Alcobendas, Madrid, Spain) for their assistance in collecting patients for the study.

### Funding

The authors received no funding for this work.

### Competing Interests

The authors declare there are no competing interests.

### Author Contributions

- Mónica Grande-Alonso conceived and designed the experiments, performed the experiments, analyzed the data, prepared figures and/or tables, authored or reviewed drafts of the paper, and approved the final draft.
- Daniel Muñoz-García analyzed the data, prepared figures and/or tables, authored or reviewed drafts of the paper, and approved the final draft.
- Ferran Cuenca-Martínez and Alfonso Gil-Martínez analyzed the data, authored or reviewed drafts of the paper, and approved the final draft.
- Laura Delgado-Sanz performed the experiments, prepared figures and/or tables, authored or reviewed drafts of the paper, and approved the final draft.
- María Prieto-Aldana performed the experiments, authored or reviewed drafts of the paper, and approved the final draft.
- Roy La Touche conceived and designed the experiments, analyzed the data, authored or reviewed drafts of the paper, and approved the final draft.

### Human Ethics

The following information was supplied relating to ethical approvals (i.e., approving body and any reference numbers):

The Ethics Committee of La Paz Universitary Hospital (PI-2567).

### Data Availability

The raw measurements are available in the Supplemental File.

### Supplemental Information

Supplemental information for this article can be found online at http://dx.doi.org/10.7717/peerj.8756#supplemental-information.

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
