# Peer review of "Relationship between healthcare seeking and pain expansion in patients with nonspecific chronic low back pain"

_PeerJ, doi:10.7717/peerj.8756_

## Round 0.1 · original submission · Major Revisions

I have had the manuscript reviewed by two people with expertise in low back pain in general, and in persistent low back pain in particular. They have raised points in common and some independent points too. The common reservations are ones I share - the study has some potentially fundamental flaws, but you may be able to overcome them with some major revisions. I want to make it clear that the revisions are major and I will ask the reviewers to respond once you submit the revised version.
I must apologise for the delay in getting this decision to you - it reflects unfortunate timing on my part and, knowing what it is like to wait a long time for manuscript reviews, you must have been getting very annoyed.

·

Basic reporting

I think the manuscript should be edited by a native English speaker. Examples of strange sentences or errors:
- However, another study have not found the same results
- Affectation?
- Flyers were placed at the university centre and also at the local community, and those patients with CLBP who were not seeking treatment, were recruited. On the other hand, those patients who have gone to a health professional due to the presence of low back pain were classified in the care seekers group. It is therefore, these patients had an intention to be treated.
- This process might a reason why in our patients seeking help had a greater expansion of…

Use consistent language: extension vs expansion.

Use consistent reference style: only surname, when all names, when et al?

Experimental design

Introduction:
- I would recommend to at least mention somewhere the newer introduced term of nociplastic pain, and how it relates to CS.
- I think the research question is not sufficiently introduced and teh relevance is not argued in the introduction.

Methods:
It would be better to restructure the methods. Cluster all information, regarding sample (now e.g. spread over the beginning, middle part and end of methods), the information regarding procedure, and then all information regarding the outcome measures.…
Add clinimetrics or references for all outcome measures.
Strange that the introduction is about central sensitization, and not/less about physical functioning etc., while in the methods outcomes are mainly on physical performance and cognitions? So again, I think the introduction does not fit with the rest of the article.
Cohen’s d effect sizes were calculated for multiple comparisons of the outcome variables. What do you mean with the “for multiple comparisons”?

Results:
Results can be shortened as everything is shown in tables.

Discussion:
I think, overall, the authors did a good job in discussing their study, although the discussion is lengthy. Readability would increase when the discussion would be shortened and restructured on some points. Cluster your paragraphs around similar constructs.
Regarding the limitations: I think more limitations need to be addressed (see earlier comments). I think the argument of the opioid system is not the most appropriate for discussing the differences in PA levels. Think the role of PA in pain deserves a better and more extensive explanation. Furthermore, I would rather consider the difference in PA also as a very interesting outcome, rather than discussing it in the light of comparability of groups? I have more problems with the age-thing! This could be more coincidence, or caused by selection bias, and might be independent from the complaints. So I think that is more confounding the study, while PA is in my opinion an interesting outcome?
For your conclusion and the rest of your article (e.g. statistics) always stick to the same consistent order of your objectives. So start with the expansion of pain, etc.

Validity of the findings

Methods:
The care seeker group: where they recruited via therapists/doctors?
More specific in- and exclusion criteria are lacking: age, comorbidities, time between seeking care and recruitment, …
Outcome measures:
- Frequency of medication: so someone taking 10 days 1 paracetamol = person taking 3 tramadols for 10 days?
- Why no hyperalgesia, allodynia, or some kinds of QST?
- Explain the difference between PPSA and WP?
Statistics:
- The Student T-test was used for the non-specific CLBP group comparisons. Why not everything with Anova?

Results:
- There were differences in age between groups, were they recruited from the same population? Could you account for this difference?
- Table 2: can you indicate the significant ones? Or add p-values?

·

Basic reporting

In the abstract and background there's a lot of speculation regarding causes of chronic low back pain and even a suggestion that CLBP can cause central sensitization, which is a theory worth exploring although completely irrelevant to the topic of this study.
What is lacking, however, is a background that provides the reader with an understanding of why this study has relevance (i.e. previous studies or underlying reasoning that have let the authors to believe that people who seek care are different from those who don't). It remains a mystery to the reader why the authors have included all the secondary analysis if the one true finding is that there is a significantly increased spreading of the pain (pain drawing) in one group compared to the two others. The reader should be presented with this information in a clear structure from background through to conclusion. To this end, a complete re-write of the manuscript appears the most obvious solution.
There are some issues with the language, which seem to confuse the authors; they seem most reliant on american spelling (e.g. line 51, 78 and 330) but use UK spelling for the esthesiometer. I suggest using only one. A general review of the written language is recommended.

In the results section the main results are hidden and only referred to as Figure 1 and Figure 2. Please provide the main results in the first or second paragraph.

Experimental design

A convenience sample is used, yet the basis for sample size selection is unclear; please provide precise data on effect measures and units used to calculate the sample size.

The research question is not clear (no hypothesis or relevant background is provided).

Please elaborate on the assumption that data was normally distributed; did you test it?

The manuscript should clearly show which results can be considered relevant to the power / sample size of the study to avoid any suspicion of type-2 errors. Group differences are analyzed with one-way ANOVA, but if the authors suggest that there is differences in the secondary variables, which could explain the group differences, why did they not use 2-way-ANOVA to assess their relationship?

Please justify that your measurements are trait, not state, dependent or in other ways justify that a single measure (at one time point) can support your conclusions (a non-seeker could seek care the day after the study; how would this influence the results)?

Validity of the findings

Results based on the sample-size calculation are not clearly separated from those of more speculative origin. This undermines the validity of the results and must be corrected. In the current format it is not possible to neither accept nor reject the findings since the outcome used to calculate power is unknown.

There are significant differences between the age of those who seek care compared to those who don't and controls. This is a major limitation and must be thoroughly discussed, if the authors believe that age cannot account for the results. Furthermore, authors should explain why 'age' is not analyzed in a one-way ANOVA since this is the way all other data is analyzed.

Finally, the conclusion should report on the primary measures only (spreading of pain).

Additional comments

It seems like a relevant finding that pain is reported to be more wide-spread among patients who seek care compared to people living with pain (and asymptomatic people). This should be the main focus of the article (the rest seems to be speculative or in need of justification methodologically). It would be interesting to know why you thought of this hypothesis in the first place (and indeed what you expected to find and why). The methods and results could be trimmed tremendously if they only account for the (exptected) hypothesis.
In the discussion there are several limitations and considerations, including clinicial relevance and least-clinically relevant difference, that needs to be addressed.

---

## Round 0.2 · Major Revisions

First and foremost, I am afraid you have suffered again from unfortunate timing at my end: the reviewers (who are in Europe) completed their task in a timely fashion but it timed with our Christmas break in Australia and then the arrival of a rather terrible bushfire season that had many of us dropping everything. However, the rain has come and I am back on deck.

Second, the reviewers appreciated the extensvie changes you made and I agree that the paper is better for it. There are three main outstanding concerns that I see as very important. Two are articulated very clearly by the reviewers. The third, also mentioned by the reviewers, concerns the elevated risk of false positives when a study undertakes multiple analyses without adequately preparing and catering for it a priori. Your explanation about the nature of false negatives and positives as set out by Cohen is clear. However, the issue as I see it for your paper is that you undertake many analyses. Each one is associated with a risk of false positive and, roughly speaking, your likelihood of a false positive increases by 5% for every analysis. I think this point is tied into the concerns of Reviewer 2. Without lodging and locking a protocol that predicts your hypotheses and primary outcomes, any matter like this becomes high risk. As such, a second researcher evaluating this paper for inclusion in a systematic review for example, would have to allocate it a high risk of bias.

It is not clear to me how you can get around this now. However, the reviewers are both supportive of your work and I think if you clearly articulate this problem in your discussion, it will suffice.

Please respond to the two other concerns from reviewers, taking particular care to address the significant problem raised by Reviewer 2 linking the outcomes with a sound rationale.

·

Basic reporting

Introduction:
I am still concerned about the introduction. The language is not always clear, and consequently I don’t get the message soon enough. It is only at the end of the introduction that it becomes clear to me where the study is going to.
Do we need all the information about proprioception, about sensitization etc.? Because of the overload of information on more irrelevant points, the flow of the introduction is not clear.

Results:
OK, improved!

Discussion:
Still some strange and long sentences, that are not always easy to read/interpret.

Suggestion to change "seek care" to care-seeking whens used as descriptor or noun.

Experimental design

OK, rebuttal and changes OK.

Validity of the findings

OK

·

Basic reporting

The study needs some corrections:
- Language review for readability (e.g. lines 88-93 and 286)
- Content review; there are several examples of confusion between causality and associations (e.g. lines 83-86 and 90-93)
- Clarification; how was fibromyalgia (excluded) differentiated from wide-spread pain (focus of the results)? If only by diagnose, was it really excluded or was it only people with known rheumatological diagnosis including FMS that were excluded? And if so, how could this influence the results?
- There are some factual mistakes in the text including the defintion of nociplastic pain (see IASP for true definition) and the role of tissue damage in chronic pain (line 58). I strongly encourage the authors to use original references and to reference them directly if they decide to re-submit.

Experimental design

The design itself seems appropriate but results focus too much on secondary outcome. In the current format this not only confuses the reader, it makes the study too weak to publish.

Twice in the study, the authors reference methods from a different study with no further description. If they used the exact same methods, please describe in an abbreviated version how the methods were the same to give the reader an impression of the data. If only some methods were re-used, please refrain from copying in a reference, and describe in full what was done in the current study.

Validity of the findings

The following results and related analysis are relevant:

"Statistically significant differences have been observed in the pain drawing, being higher in care seekers (Figure 1 and Figure 2). One-way ANOVA revealed significant differences for pain drawings (F=14.49, p < .001, η p 2=.250). The post hoc analysis showed statistically significant inter-group differences between patients with non-specific CLBP who do not seek care and patients with non-specific CLBP who seek care with a medium effect size (p = .009, d=-0.64). As well as, between asymptomatic controls and patients with non-specific CLBP who seek care with a large effect size (p < .001, d=-1.19)."

There is a strong need for the authors to address (discuss) the possible influence of the differences between CLBP-groups (pain intensity, age and educational level), including a discussion of the risk for type-1 and type-2 errors.

To the extend that the secondary data relates directly to the primary data, and they provide novel or relevant information, they can be briefly mentioned with an emphasis on the speculative nature. If not, all analysis, results and discussion of secondary outcomes should be taken out of the manuscript.

Additional comments

Speading of pain is a clinically relevant observation used in most areas of clinical reasoning. As such the study takes a good perspective although it is very difficult to understand how this should have any academically relevant association towards who seeks care? This innate problem in the study seems to be main area of concern in the current manuscript since it leads the authors to lose focus. If they believe that there is an academically relevant association between seeking healthcare and the spreading of pain, it remains to be revealed in the manuscript.
The speculative associations of secondary outcomes to the primary only confuses the message and should be left out completely. Two-point discrimination, which has well-documented associations to pain spreading has already been described in patients with CLBP, but the current study does not seem to add anything to the existing evidence, which it could be left out.

I think the authors should consider discussing Landmark et al. (https://journals.lww.com/pain/Abstract/2019/09000/Development_and_course_of_chronic_widespread_pain_.8.aspx) in relation to their background as well as discussion.

---

## Round 0.3 · accepted · Accept

Thank you for your thorough response to the reviewers' outstanding comments. I think your consideration of the limitation in the discussion is satisfactory. Congratulations.